# A Special Amino-Acid Formula Tailored to Boosting Cell Respiration Prevents Mitochondrial Dysfunction and Oxidative Stress Caused by Doxorubicin in Mouse Cardiomyocytes

**DOI:** 10.3390/nu12020282

**Published:** 2020-01-21

**Authors:** Laura Tedesco, Fabio Rossi, Maurizio Ragni, Chiara Ruocco, Dario Brunetti, Michele O. Carruba, Yvan Torrente, Alessandra Valerio, Enzo Nisoli

**Affiliations:** 1Center for Study and Research on Obesity, Department of Medical Biotechnology and Translational Medicine, University of Milan, 20129 Milan, Italy; 2Department of Pathophysiology and Transplantation, University of Milan, Fondazione IRCCS Ca’ Granda Ospedale Maggiore Policlinico, Centro Dino Ferrari, 20122 Milan, Italy; 3Department of Molecular and Translational Medicine, University of Brescia, 25121 Brescia, Italy

**Keywords:** branched-chain amino acids, cardiomyocytes, doxorubicin, endothelial nitric oxide synthase, Krüppel-like factor 15, mechanistic/mammalian target of rapamycin, mitochondria, oxidative stress, peroxisome proliferator-activated receptor γ coactivator 1 α, tricarboxylic acid cycle

## Abstract

Anthracycline anticancer drugs, such as doxorubicin (DOX), can induce cardiotoxicity supposed to be related to mitochondrial damage. We have recently demonstrated that a branched-chain amino acid (BCAA)-enriched mixture (BCAAem), supplemented with drinking water to middle-aged mice, was able to promote mitochondrial biogenesis in cardiac and skeletal muscle. To maximally favor and increase oxidative metabolism and mitochondrial function, here we tested a new original formula, composed of essential amino acids, tricarboxylic acid cycle precursors and co-factors (named α5), in HL-1 cardiomyocytes and mice treated with DOX. We measured mitochondrial biogenesis, oxidative stress, and BCAA catabolic pathway. Moreover, the molecular relevance of endothelial nitric oxide synthase (eNOS) and mechanistic/mammalian target of rapamycin complex 1 (mTORC1) was studied in both cardiac tissue and HL-1 cardiomyocytes. Finally, the role of Krüppel-like factor 15 (KLF15), a critical transcriptional regulator of BCAA oxidation and eNOS-mTORC1 signal, was investigated. Our results demonstrate that the α5 mixture prevents the DOX-dependent mitochondrial damage and oxidative stress better than the previous BCAAem, implying a KLF15/eNOS/mTORC1 signaling axis. These results could be relevant for the prevention of cardiotoxicity in the DOX-treated patients.

## 1. Introduction

Anthracyclines, such as doxorubicin (DOX), are widely used and highly successful anticancer chemotherapeutics [1]. Unfortunately, DOX administration results in dose-dependent side effects to non-cancer tissues, including the development of cardiomyopathy, in addition to dyspnea, exercise intolerance, hepatotoxicity, and nephropathy [2]. The risk of cardiotoxicity is one of the greatest limiting factors to the clinical use of this drug, resulting in both acute and chronic cardiovascular events. Acute cardiac toxicity of DOX can develop within minutes to days after administration and normally is characterized by hypotension, arrhythmia, and most importantly left ventricular failure [1,3,4]. Although the molecular mechanisms of these side effects are not fully understood, also because DOX affects many different intracellular processes, increasing evidence suggests that the primary mediator of cardiac damage by DOX is oxidative stress, with increased reactive oxygen species (ROS)-dependent lipid peroxidation and reduced levels of antioxidants and sulfhydryl groups [1,5]. In particular, mitochondrial dysfunction plays a critical role, because DOX accumulates in mitochondria, binding to the inner membrane lipid cardiolipin and complexing with respiratory chain proteins [5,6,7]. The high production of ROS then causes mitochondrial impairment, with reduced ATP synthesis, and apoptosis of cardiac cells. Moreover, DOX decreases mitochondrial biogenesis as a result of binding to topoisomerase IIβ and complexing to the promoters of *Ppargc1a* and *Ppargc1b*, which is likely responsible for blocking transcription of mitochondrial genes [8]. Upregulation of manganese superoxide dismutase (MnSOD or SOD2)—that transforms toxic superoxide into hydrogen peroxide (H_2_O_2_) and diatomic oxygen—protects cardiac cells exposed to DOX acting as a free radical scavenger in mitochondria [9]. Moreover, overexpression of glutathione peroxidase 1 (Gpx1), a cytosolic and mitochondrial enzyme that reduces H_2_O_2_ and fatty acid hydroperoxides, prevents cardiotoxicity, mitochondrial dysfunction and inhibition of respiratory complex 1 activity induced by DOX [10].

Calorie restriction (CR) prolongs lifespan by preventing inflammatory diseases, cardiovascular disorders, and cancer. Increasing evidence emphasizes the relevance of proteins involved in mitochondrial pathways to explain the health effects of CR, including peroxisome proliferator-activated receptor γ coactivator 1 α (PGC-1α)—the master regulator of mitochondrial biogenesis [11]. Additionally, CR has been found to reduce cardiac dysfunction in rats treated with DOX [12]. Hence, one could hypothesize to combine the CR regimen with DOX treatment protocols with the aim of mitigating cardiotoxicity. However, it is unlikely that such a dietary regimen could be adopted for patients affected by cancer; many researchers are focusing in fact on the development of CR mimetic compounds that provide the benefits of dietary restriction without reducing calorie intake [13].

Recently, we have demonstrated that chronic (3 months) dietary supplementation with a branched-chain amino acid-enriched mixture (BCAAem) promoted mitochondrial biogenesis in cardiac and skeletal muscle of middle-aged mice, with marked reduction of oxidative stress and lifespan extension [14]. These effects were mediated by endothelial nitric oxide synthase (eNOS) and mechanistic/mammalian target of rapamycin (mTOR) signaling pathways [14]. To maximally favor and increase oxidative metabolism and mitochondrial function, here we tested an original formula composed of essential amino acids, tricarboxylic acid cycle intermediates, and co-factors (named α5) and explored its effect in HL-1 cardiomyocytes and mice treated with DOX. As compared to BCAAem, α5 supplementation was more effective in promoting protective effects on the DOX-induced mitochondrial dysfunction. Thus, we extended these results *in vivo* in young mice exposed to acute DOX treatment. Our findings confirm the occurrence of mitochondrial dysfunction after acute DOX, demonstrate marked defensive validity of short-term (10 days) supplementation with the new α5 formula, and suggest that Krüppel-like factor 15 (KLF15), eNOS, and mTOR signaling pathways are crucially involved in the action of this defender.

## 2. Materials and Methods

### 2.1. Cell Cultures and Treatments

The paper follows the rules of the Declaration of Helsinki. HL-1 cardiomyocytes were obtained from W.C. Claycomb (Cat# SCC065, Millipore, Milan, Italy) and plated in fibronectin/gelatin-coated flasks, grown to 70–80% confluence in Claycomb medium (Sigma-Aldrich, Milan, Italy) supplemented with 100 μM norepinephrine (from a 10 mM norepinephrine [Sigma-Aldrich] stock solution dissolved in 30 mM L-ascorbic acid [Sigma-Aldrich]), 2 mM L-glutamine, 100 U/mL penicillin, 100 μg/mL streptomycin and 10% fetal bovine serum (FBS, Sigma-Aldrich) [15,16]. MCF-7 human breast cancer cell line was obtained from P. Limonta (Pharmacological and Biomolecular Sciences, University of Milan, Milan, Italy) and cultured in pH 7.4 DMEM, containing streptomycin (100 U/mL), penicillin (200 mg/mL), and gentamicin (50 mg/mL), and supplemented with 10% FBS. Both cell types were treated with 1% BCAAem or α5 for 48 h and 1 μM DOX for 16 h (Figure 1). The detailed composition percentages of mixtures are shown in Table 1.

For the study of phospho-proteins, HL-1 cells were treated with 1% α5 for 2 h and 1 μM DOX for the last 60 min. For *Klf15*, *eNOS*, and *Raptor* knockdown experiments, HL-1 cells were transfected with 50–100 nM *Klf15*, *eNOS*, and *Raptor* siRNA SMARTpool (Dharmacon; Lafayette, CO, USA) or siGENOME nontargeting siRNA using Dharmafect 1 transfection reagent. After 24 h transfection, cells were treated with 1% α5 for 24 h and 1 μM DOX for 16 h. Transfection efficacy was determined with siGLO-RISC-free non-targeting siRNA and siRNA uptake by fluorescence detection (absorbance/emission 557/570). Proteins were then extracted for western blotting analysis.

### 2.2. Animals and Treatments

The experimental protocol used was approved by the Institutional Ethical Committee of Milan University (n. 16/09) and complied with the National Animal Protection Guidelines. Forty male C57BL6/J mice (9 weeks-old) were housed separately in clean polypropylene cages and divided into four groups (Figure 2): (1) the control group (CTRL, *n* = 10 mice) fed with standard diet (4.3 kcal % fat, 18.8 kcal % protein, 76.9 kcal % carbohydrate; Laboratorio Dottori Piccioni, Gessate, Italy) and receiving a single i.p. saline injection (vehicle); (2) the α5 group (*n* = 10 mice) fed with standard diet and α5 supplementation (1.5 mg/g body weight/day in drinking water) receiving a single i.p. saline injection (vehicle). α5 mixture was dissolved in tap water, after calculating the average daily drinking volume 2 weeks before the start of treatment and stored at 4 °C before daily administration; (3) the DOX group (*n* = 10 mice) fed with standard diet and receiving i.p. DOX (Doxo-HCl from Sigma-Aldrich) injection at 20 mg/kg, a dose that had been shown cardiotoxic [17,18,19]; and 4) the DOX plus α5 group (*n* = 10 mice) fed with standard diet and receiving i.p. 20 mg/kg DOX injection plus α5 supplementation (1.5 mg/g body weight/day in drinking water). α5 supplementation was performed for 10 days, with a 12 h light/12 h dark cycle at 22 °C in a quiet, temperature- and humidity-controlled room; single dosing of DOX was performed on the third day before the end of α5 treatment (Figure 2).

Drinking volume, food intake, and body weight were checked twice weekly. At the end of the treatment period, mice were sacrificed by cervical dislocation and hearts quickly removed and freshly used (for oxygen consumption analysis) or frozen in liquid nitrogen and stored at −80 °C (for mtDNA, mRNA, and protein level measurement, in addition to citrate synthase activity analysis).

### 2.3. Quantitative RT-PCR Analysis

Quantitative RT-PCRs were performed as previously described [16,20] with the iQ SybrGreenI SuperMix (Bio-Rad; Segrate, Italy) on an iCycler iQ real-time PCR detection system (Bio-Rad). Briefly, RNA was isolated from left ventricle using the RNeasy Tissue Mini Kit (Qiagen, Segrate, Italy) or from HL-1 cells using the RNeasy Mini Kit (Qiagen). cDNA was synthesized using the iScript cDNA Synthesis Kit (Bio-Rad Laboratories). Primers were designed with Beacon Designer 2.6 software from Premier Biosoft International (Palo Alto, CA, USA) and are shown in Table 2.

The cycle number at which the various transcripts were detectable (threshold cycle, CT) was compared to that of TBP, referred to as ΔCT. The relative gene level was expressed as 2^-(ΔΔCT)^, in which ΔΔCT equals the ΔCT of DOX-, α5-, or DOX plus α5-treated mice (or treated HL-1 cells) minus the ΔCT of the control mice (or untreated HL-1 cells).

### 2.4. Western Blot Analysis

Protein extracts were obtained from left ventricle with T-PER mammalian protein extraction reagent (Pierce, ThermoScientific, Rockford, IL, USA) or from HL-1 cells in M-PER mammalian protein extraction reagent (Pierce), as indicated by the manufacturer, in the presence of 1 mM NaVO_4_, 10 mM NaF and a cocktail of protease inhibitors (Sigma-Aldrich, Milan, Italy). Protein content was determined by the bicinchoninic acid protein assay (BCA, Pierce, Euroclone, Milan, Italy), and 50 µg of the protein extract was separated by SDS-PAGE under reducing conditions. The separated proteins were then electrophoretically transferred to a nitrocellulose membrane (Bio-Rad Laboratories, Segrate, Italy) [16,21]. Proteins of interest were detected with specific antibodies: anti-COX IV (cytochrome c oxidase subunit IV, Cell Signaling Technology Cat #4844, Euroclone, Milan, Italy), anti-cyt c (cytochrome complex, Cell Signaling Technology Cat #4280), anti-PGC-1α (proliferator-activated receptor γ coactivator 1α, Cell Signaling Technology Cat #2178), anti-phospho-AKT (Ser473) (Cell Signaling Technology Cat #4060), anti-AKT (Cell Signaling Technology Cat #4685), anti-phospho-eNOS (Ser1177) (phospho-endothelial nitric oxide synthase, Cell Signaling Technology Cat #9571), anti-eNOS (Cell Signaling Technology Cat #9572), anti-phospho-S6 (Ser235/236) (phospho-S6, Cell Signaling Technology Cat #4858), anti-S6 (Cell Signaling Technology Cat #2217), anti-phospho-mTOR (Ser2481) (Cell Signaling Technology Cat#2974), anti-mTOR (Cell Signaling Technology Cat#2972), anti-phospho-BCKDH (Ser293) (Abcam Cat #200577), anti-BCKDH (Abcam Cat #138460), and anti-GAPDH (1:1000, Cell Signaling Technology Cat #2118) at 1:1000 dilution each. After the visualization of phospho-eNOS, phospho-AKT, phospho-mTOR, phospho-S6, and anti-phospho-BCKDH the filters were stripped with the Restore^TM^ western blot stripping buffer (Euroclone, Milan, Italy) and further used for the visualization of total eNOS, total AKT, total mTOR, total S6, or total BCKDH. Immunostaining was performed using horseradish peroxidase-conjugated anti-rabbit or anti-mouse immunoglobulin for 1 h at room temperature. The protein was detected using SuperSignal substrate (Pierce, Euroclone, Milan, Italy) and quantified by densitometry with ImageJ image analysis software.

### 2.5. Mitochondrial DNA

For mtDNA analysis, total DNA was extracted with a QIAamp DNA Extraction Kit (Qiagen). mtDNA was amplified using primers specific for the mtDNA and normalized to the 18S gene (Table 2) [14]. The mtDNA content was determined using qRT-PCR by measuring the threshold cycle ratio (ΔCT) of the mtDNA gene *vs.* that of a nuclear encoded gene (18S) in left ventricle of DOX-, α5- or DOX plus α5-treated mice, in addition to control untreated mice [22,23].

### 2.6. Citrate Synthase Activity

The activity was measured spectrophotometrically at 412 nm at 30 °C in left ventricle extracts [24,25]. Tissue was added to buffer containing 0.1 mM 5,5-dithio-bis-(2-nitrobenzoic) acid, 0.5 mM oxaloacetate, 50 μM EDTA, 0.31 mM acetyl CoA, 5 mM triethanolamine hydrochloride, and 0.1 M Tris-HCl (pH 8.1). Citrate synthase activity was expressed as nmol citrate produced per min per mg of protein. The data were normalized to total protein content, which was determined by the bicinchoninic acid assay, as reported above.

### 2.7. Oxygen Consumption

Oxygen consumption was measured as described [26,27]. Mitochondria were isolated from left ventricle of control and treated-mice. Samples were analyzed at 37 °C in a gas-tight vessel equipped with a Clark-type oxygen electrode (Rank Brothers Ltd., Bottisham, Cambridge, England) connected to a chart recorder. The oxygen electrode was calibrated assuming the concentration of oxygen in the incubation medium as 200 μmol/L at 37 °C.

### 2.8. Mitochondrial Oxidative Stress

To investigate the mitochondrial oxidative stress, mitochondria were isolated using the Qproteome Mitochondria isolation kit (Qiagen). Mitochondrial H_2_O_2_ release was measured in the presence of horseradish peroxidase (HRP), using the Amplex Red Hydrogen Peroxide/Peroxidase Assay kit (Molecular Probes, Monza, Italy). Fluorometric measures were made using a Fusion Universal Microplate Analyzer (Packard/PerkinElmer, Milan, Italy) with excitation filter at 550 nm and emission filter at 590 nm. H_2_O_2_ production, calculated from a standard curve, was expressed as nmol/ min/mg protein as described [14]. Moreover, mitochondrial aconitase and SOD activity were measured as previously described [28]. Briefly, the formation of NADPH was followed spectrophotometrically (340 nm) at 25 °C (for aconitase activity), while SOD activity was measured with the Superoxide Dismutase Assay Kit (Calbiochem, Milan, Italy). One unit of SOD activity was defined as the amount of the enzyme needed to exhibit 50% dismutation of the superoxide radical. Finally, the oxidative damage of DNA was measured as a further marker of oxidative stress. The highly sensitive 8-hydroxy-2’-deoxyguanosine (8-OHdG) Check ELISA Kit (JaICA, Hamamatsu, Japan) was used [29]. Measurements were carried out in accordance with the manufacturer’s protocol. Total DNA was extracted using QIampDNAMini Kit (Qiagen) and digested with nuclease P1 and alkaline phosphatase (Sigma). Quality and quantity of DNA were confirmed by a NanoDrop ND-1000 spectrophotometry analysis. Absorbance of the ELISA reaction product was determined spectrophotometrically using 450 nm as the primary wave.

### 2.9. Viability Assay

HL-1 cell viability was assessed by MTT [3-(4,5-dimethylthiazol-2-yl)-2,5-diphenyltetrazolium bromide] reagent (Sigma). HL-1 were seeded into 96-well culture plates at density 20,000 cells/well (100 μL). The purple formazan crystals were dissolved in 5% SDS/0.1 M HCl (100 μL/well), and the absorbance was recorded on a microplate reader (EL×800, BioTek Instruments, Winooski, VT, USA) at a wavelength of 570 nm. Each test was repeated at least four times in quadruplicates.

### 2.10. Acid Phosphatase Assay

To quantify MCF7 cell growth, acid phosphatase assay was used as described [30]. Briefly, MCF7 cells were placed in 96-well plates at 5000 to 20,000 cells per well density and treated with 1% α5 (for 48 h) and 1 μM DOX (for 16 h). Culture medium was removed and each well washed once with phosphate-buffered saline (PBS, pH 7.2), and added with 100 µL buffer containing 0.1 M sodium acetate (pH 5.0), 0.1% Titon X-100, and 5 mM p-nitrophenyl phosphatase (pNPP). Then, plates were placed in a 37 °C incubator for 2 h. The reaction was stopped with the addition of 10 µL 1 N NaOH, and color development was assessed at 405 nm. Non-enzymatic pNPP hydrolysis was measured in wells without cells.

### 2.11. Statistical Analysis and Data Presentation

Statistical analysis was performed with a one-way ANOVA followed by Student-Newman-Keuls’ test or Student’s *t*-test. Data are presented as the means ± standard deviations (SD), unless otherwise specified. A statistically significant difference was accepted at *p* < 0.05.

## 3. Results

### 3.1. Specific Amino-Acid Mixtures Prevent Mitochondrial Dysfunction in HL-1 Cardiomyocytes Acutely Exposed to DOX

To protect cardiomyocytes against DOX toxicity, we propose to correct the impaired mitochondrial function and oxidative stress in HL-1 cells exposed to the chemotherapeutic. An optimal combination of relevant metabolic precursors capable of maximally increase oxidative metabolism in HL-1 cardiomyocytes was evaluated. Specifically, we tested the effect of different combinations of essential amino acids, tricarboxylic acid cycle (TCA) precursors, and cofactors on differentiating HL-1 cells (Table 1). For this screening, we treated for 48 h differentiating cardiomyocytes with 1) the specific mixture of essential amino acids we previously described (i.e., BCAAem) or 2) a new specific mixture of essential amino acids, TCA precursors, and cofactors (referred to as α5) (see Materials and Methods), with or without DOX (Figure 1). While α5 (1% w/v) increased the mRNA levels of mitochondrial biogenesis markers in HL-1 cells, including *PGC-1α*, *nuclear respiratory factor 1* (NRF1), *mitochondrial transcription factor A* (Tfam), *cytochrome c* (cyt c), and *cytochrome c oxidase complex IV* (COX IV) over the basal value, only PGC-1α expression was statistically increased by BCAAem (Figure 3A). Conversely, the expression of these genes was decreased, as expected, when cardiomyocytes were exposed to 1 μM DOX for 16 h (Figure 3A). Notably, α5 supplementation prevented this DOX toxicity with a complete rescue of the gene expression (Figure 3A). Except for cyt c and COX IV, BCAAem was unable to reverse the DOX effect with statistical significance (Figure 3A). Hence, we chose to use the α5 mixture hereafter. Notably, Appendix A shows that TCA intermediates—citric acid, succinic acid, and malic acid supplemented individually or all together at the same concentrations in α5—were unable to prevent the DOX-induced reduction of mitochondrial biogenesis as well as to change the mitochondrial gene expression when supplemented alone (data not shown). The protective capacity of α5 supplementation on mitochondrial health was also evident at the protein level (e.g., COX IV and, with a trend, cyt c) (Figure 3B).

Additionally, the reduction of citrate synthase activity promoted by DOX exposure—which is an index of reduced mitochondrial mass and function—was fully antagonized by α5 supplementation, which raised the enzyme activity also when added alone to the HL-1 cells (Figure 3C). These healthy effects of α5 on the DOX-induced mitochondrial damage were further confirmed by the reduction of oxidative stress. When compared to untreated cells, H_2_O_2_ release (an index of mitochondrial superoxide anion production) was markedly increased by DOX treatment; α5 supplementation prevented this effect (Figure 4A). α5 reduced H_2_O_2_ release also when added alone. Accordingly, measurements of mitochondrial ROS production (as assessed by the basal/total aconitase activity ratio) and ability to eliminate superoxides through SOD activity demonstrated that α5 supplementation prevented the oxidative stress induced by DOX, with beneficial effects also when added alone (Figure 4B,C). Because oxidative stress ignites the anti-ROS defense system, we investigated the expression of anti-ROS enzymes. In particular, the expression of *glutathione peroxidase 1* (GPX1) and *superoxide dismutase 1* (SOD1) genes was increased in DOX-treated compared to untreated HL-1 cells (Figure 4D), consistently with the increase of ROS production [1,7].

This was confirmed by the higher amount of 8-hydroxy-2’-deoxyguanosine (8-OHdG), a marker of oxidative DNA damage, in cells exposed to DOX (Figure 4E). The α5 supplementation in DOX-treated cells was able to counteract ROS production as evidenced by reduced expression of both *SOD1* and *GPX1* genes (Figure 4D) and by restored 8-OHdG production to amount observed in the untreated cells (Figure 4E). Figure 4D shows moreover that α5 is more effective than BCAAem in the anti-ROS protection. Together, these results support the notion that α5 mixture can prevent mitochondrial damage induced by acute exposure to DOX in HL-1 cardiomyocytes. Notably, this might be of relevant impact on cell survival. The DOX-induced death of HL-1 cells in fact was prevented by α5 mixture when supplemented together (Appendix A). No effect was evident on the cell survival when HL-1 cells were exposed to α5 alone (Appendix A).

### 3.2. α5 Mixture Prevents Mitochondrial Dysfunction in Heart of DOX-Treated Mice

To confirm the *in vitro* results, an acute *in vivo* DOX treatment was performed as described [19]. On the third day before the end of the α5 treatment—which was performed for 10 days—a single i.p. injection of 20 mg/kg DOX was done (Figure 2). Table 3 shows that DOX treatment significantly decreased body weight and heart weight as compared to the control group as expected [31], and this might depend on its marked anorexigenic effect. α5 was unable to modify body weight, heart weight, and food intake both when supplemented alone and with DOX (Table 3). On the contrary, while water consumption in DOX-treated mice was unchanged compared to control animals, α5 increased water intake either when supplemented alone or with DOX.

The mRNA levels of mitochondrial biogenesis genes were reduced in the left ventricle of DOX-treated mice as compared to the control group, confirming the previous results (Figure 5A) [7]. Beyond increasing PGC-1α, cyt c, and COX IV mRNA levels when supplemented alone, the α5 mixture was able to prevent significantly the DOX-induced reduction of PGC-1α, Tfam, and cyt c (Figure 5A). Mitochondrial mass and function, measured as mitochondrial DNA (mtDNA) amount, respiratory proteins (particularly COX IV), and citrate synthase activity were lower in DOX-treated than in saline-treated mice, and these reductions were prevented when the mice were supplemented with α5 (Figure 5B–D). Moreover, the mitochondrial respiratory function was investigated by measuring oxygen consumption rate (OCR) with Clark’s electrode in mitocondria isolated from the left ventricle of the diverse treatment groups. Figure 5E shows that DOX injection decreased OCR, while α5 supplementation fully conserved the mitochondrial respiration of the DOX-treated mice. The amino acid mixture increased OCR also when supplemented alone (Figure 5E). Finally, the results reported in Figure 5F confirmed in *ex vivo* samples that the DOX augmented the expression of ROS defense enzymes, including SOD1, SOD2, catalase, and GPX1, as reported previously [7], while the α5 supplementation blocked almost completely this effect. No changes were seen in mice treated with α5 alone (Figure 5F). Collectively, these findings suggest that α5 supplementation prevents the DOX-induced mitochondrial toxicity in heart, at least in part, by promoting mitochondrial biogenesis and function, as well as by reducing oxidative stress.

### 3.3. Different Signaling Pathways are Implicated in the Protective Effects of α5 Supplementation

Because we and others found that specific amino-acid combinations improve mitochondrial homeostasis in pathological conditions through eNOS and mTORC1 activity [14,25,32], we analyzed these signaling pathways in the left ventricle of mice treated with DOX and α5. Figure 6A,B show that DOX-injection decreased eNOS expression, yet only partially eNOS activity (i.e., (Ser1177) phospho-eNOS to total eNOS ratio). Notably, the α5 supplementation fully neutralized these effects. When supplemented alone the α5 mixture was able to increase eNOS expression (Figure 6A). Given that the eNOS-dependent nitric oxide (NO) production is in turn known to regulate the mTOR complex 1 (mTORC1) signaling pathway, and that mTORC1 activity is necessary and sufficient for (Ser1177)-eNOS phosphorylation in different cell types [14,29,33,34,35], we measured phosphorylation of S6 as a downstream marker of the mTORC1 activation in our model. DOX treatment decreased (Ser235/236) phospho-S6 to total S6 ratio in the left ventricle when compared to saline injection, while the α5 supplementation at least partially prevented this effect (Figure 6C). No significant changes were observed with α5 mixture when supplemented alone. Diverse physiological stimuli modulate mTORC1 signaling by site-specific mTOR phosphorylation. Here we focused on mTORC1-associated Ser2481 mTOR autophosphorylation—the amino acid-dependent activation of mTORC1 signaling that monitors mTORC1 intrinsic catalytic activity *in vivo* [36]. Our data reveal a functional role for Ser2481 mTOR phosphorylation in DOX and α5 action, in which DOX reduced and α5 rescued this reduction of the selective mTOR phosphorylation in heart (Figure 6D). mTOR phosphorylation was increased also when α5 was added alone. Together, our results suggested that both eNOS and mTORC1 may play a role in the effects of DOX and α5 treatment.

Emphasizing this hypothesis, Sestrin2 has been recently proposed as a leucine sensor for the mTORC1 pathway, since high Sestrin2 levels inhibit mTORC1 activity when intracellular leucine concentrations are low [37]. Conversely, either low Sestrin2 levels or high intracellular leucine concentrations—displacing Sestrin2 from the mTORC1 inhibitor GATOR2—promote mTORC1 activation [37]. Given that the α5 formula contains high leucine quantity, we measured Sestrin2 expression in our conditions. As expected from our mTORC1 results and confirming Li et al.’s recent findings [38], DOX treatment markedly increased Sestrin2 mRNA in the left ventricle (Figure 6E). Notably, α5 fully prevented this DOX-induced effect, although unable to change Sestrin2 expression when supplemented alone (Figure 6E).

Similarly, Krüppel-like factor 15 (KLF15) has recently emerged as a critical transcriptional regulator of amino acid metabolism, particularly BCAA catabolism (i.e., the BCAA oxidation with production of acetyl-CoA and succinyl-CoA, two TCA intermediates), especially in the heart [39], as well as an inducer of eNOS expression in endothelial cells [40]. Thus, we studied the KLF15 expression in the left ventricle of our different treatment groups. *Klf15* mRNA levels were significantly reduced by DOX treatment, and this reduction was fully prevented by the α5 supplementation (Figure 6F). No significant changes were observed when α5 was supplemented alone (Figure 6F). Given the expression or activity of mitochondrial BCAA catabolic enzymes, such as the branched-chain α-keto acid dehydrogenase (BCKDH) complex, in the cardiac tissue is reduced in *Klf15*-null mice and heart failure [41,42], we investigated the BCKDH protein transcript, in addition to the BCKDH phosphorylation in the DOX-treated animals. When phosphorylated in fact BCKDH is inactive, while conversely when dephosphorylated it is active [43,44]. Notably, the (Ser293)-BCKDH phosphorylation was unchanged in DOX-treated compared to saline-treated mice, while α5 supplementation markedly reduced it in the presence of DOX, with an inhibitory trend effect also when supplemented alone (Figure 6G). Together, these results suggested that *in vivo* protection of mitochondrial homeostasis in cardiac tissue by α5 supplementation might be related to a KLF15/eNOS/mTORC1 signaling axis, entailing the BCAA catabolism.

To investigate more this hypothesis and to verify our findings specifically in cardiomyocytes, we sought to assess the impact of KLF15, eNOS, and mTORC1 on mitochondrial homeostasis in HL-1 cells exposed to DOX and α5. First, although eNOS gene expression was only slightly decreased by 1 µM DOX exposure, (Ser1177)-eNOS phosphorylation was markedly reduced (Figure 7 A,B). α5 (1% w/v) fully antagonized both DOX inhibitory effects, massively upregulating eNOS expression also when supplemented alone (Figure 5A,B). Second, the (Ser235/236)-S6 phosphorylation was markedly decreased by DOX, and this reduction was fully antagonized by α5 (Figure 7C). Conversely, the amino-acid mixture was ineffective when added alone. In accordance to its role as mTORC1 controller, Sestrin2 expression was markedly increased by DOX, and this increase was fully antagonized by the α5 supplementation (Figure 7D). The amino-acid mixture was ineffective when added alone. Third, based on the modulatory role of Akt, also known as protein kinase B (PKB), on both eNOS and mTORC1 activation [35], we studied also the (Ser 473)-Akt phosphorylation in HL-1 cells treated with DOX and α5. DOX markedly reduced Akt phosphorylation, while α5 prevented this effect (Figure 7E). Since the Akt/PKB signaling pathway was recently found to mediate the effect of BCAAs on the KLF15 expression [45,46], we investigated the role of this transcription regulator in the protective impact of α5 supplement on the DOX-mediated mitochondrial damage. Notably, the KLF15 expression was reduced both at mRNA and protein levels when HL-1 cells were exposed to DOX, and α5 fully antagonized this effect (Figure 7F). Only a slight, not statistically significant increase of KLF15 expression was seen in HL-1 cardiomyocytes exposed to α5 alone (Figure 7F).

The HL-1 cells were then transfected with either specific *Klf15* siRNA or nontargeting siRNA. The silencing efficacy was measured both at mRNA and protein levels (Appendix A).

DOX exposure significantly decreased both PGC-1α and COX IV protein in *Klf15* siRNA-transfected or otherwise untreated cells (Figure 7G). Conversely, the *Klf15* knockdown markedly reduced PGC1-α and COX IV levels *per se* and abolished the ability of α5 supplementation to rescue the protein levels when administered with DOX (Figure 7G). Moreover, the *Klf15* knockdown markedly reduced the ability of α5 to promote PGC1-α and COX IV expression when α5 was added alone (Figure 7G). Noteworthy, the *Klf15* knockdown massively impaired the ability of α5 to restore the DOX-reduced (Ser235/236)-S6 phosphorylation (Figure 7H). Finally, while (Ser293)-BCKDH phosphorylation was not different between DOX-treated and untreated control HL-1 cardiomyocytes, p-BCKDH fully disappeared when the cells were exposed to α5, with or without DOX (Figure 7H). *Klf15* knockdown partially prevented the inhibitory effect of α5 on (Ser293)-BCKDH phosphorylation both in DOX-treated and untreated cells, while DOX was unable to change phosphorylation when added alone (Figure 7H).

Similarly, silencing of *eNOS* and *Raptor*—one of the scaffold proteins of mTORC1 (regulatory-associated protein of mTOR)—with specific siRNAs reduced PGC1-α and COX IV levels *per se* and abolished the ability of α5 supplementation to rescue the reduced protein levels when administered with DOX (Appendix A). Both *eNOS* and *Raptor* knockdown did not change PGC1-α and COX IV expression when α5 was added alone (Appendix A). The efficacy of specific silencing was evaluated by RT-PCR and Western blot; nearly 70% and 60% down-regulation of eNOS and mTORC1 were obtained with *eNOS* and *Raptor* siRNA, respectively (Appendix A). Overall, these findings suggest that the protective effects of α5 mixture on DOX-mediated mitochondrial dysfunction in cardiomyocytes could be related to a KLF15/eNOS/mTORC1 signaling axis and BCAA catabolism.

## 4. Discussion

Although the anthracycline DOX (trade name Adriamycin) is a highly effective and frequently used antineoplastic drug since its introduction in the 1960s, it causes a dose-related cardiotoxicity that can lead to severe heart failure. If the heart has been damaged by DOX, treatment options are few. Typically, DOX-induced cardiomyopathy and heart failure are refractory to conventional therapy [2]. Increasing efforts to predict which patients will be affected by DOX cardiotoxicity—including analysis of the patient-specific human induced pluripotent stem cell-derived cardiomyocytes—and to appropriately prevent this risk, encompassing drugs, such as iron-chelating agents, angiotensin-converting enzyme inhibitors, β-blockers, antioxidants, and natural products or food supplements, have been proposed inconclusively [47,48]. Here, we have compared the efficacy of α5, a new formula of essential amino acids, TCA intermediates, and co-factors to BCAAem—an amino-acid mixture able to promote mitochondrial biogenesis and anti-ROS protection in cardiac and skeletal muscle of middle-aged mice [14]—in preventing the DOX-induced mitochondrial damage in cultured HL-1 cardiomyocytes. α5 was statistically more efficient than BCAAem to counteract the DOX-induced deficit of mitochondrial biogenesis markers. We did not investigate whether α5 supplementation was able to change the amino acid and TCA intermediate levels both in cardiomyocyte and heart tissue; this could add relevant insights on the mechanism of action of our original formula.

Further studies better aimed at quantifying metabolic changes in the heart will integrate the biochemical and functional data reported in the present study. Examining mechanistically our findings in cardiomyocytes and extending them *in vivo*, we found that the dietary supplementation of α5 activates a KLF15/Akt/eNOS/mTORC1 signaling axis (Figure 8). These results also do not exclude a role of mTORC2, a signaling pathway that seems to be essential for normal cardiac development and for the maintenance of postnatal cardiac structure and function—especially to adapt to stress conditions [49]. However, mTORC2 has not been until now implied, at our knowledge, in the DOX cardiotoxicity. Moreover, the present findings on KLF15 and Sestrin2 similarly cast doubt on the relevance of mTORC2 signaling in this experimental model. One of the most relevant findings of the present work is the massive downregulation of PGC-1α and COX-IV protein—that are mitochondrial biogenesis markers—induced by the KLF15 silencing in HL-1 cardiomyocytes. The Kruppel-like factors (KLF) family of Cys^2^/His^2^ zinc-finger transcriptional regulators control many aspects of cardiomyocyte and mitochondrial structure and function [39]. Importantly, KLF15 regulates Wnt/β-catenin transcription and controls cardiac progenitor cell fate in the postnatal heart, suggesting a role in cardiac development [50]. Moreover, grossly enlarged mitochondria (or megamitochondria) were recently found in heart of systemic and cardiac-specific *Klf15*-null mice, with clear evidence that deletion of *Klf15* interferes with mitochondrial fission, whereas fusion appears to be unaffected. This was accompanied by a specific impairment of cardiac function upon fasting [51].

We have shown here that the *Klf15*-, *eNOS-*, and *Raptor*-silenced cardiomyocytes were unresponsive to α5, which was unable to promote mitochondrial biogenesis *per se* and to protect the mitochondrial damage when supplemented with DOX. Moreover, we found—for the first time at our best knowledge—that after a 72-h treatment with DOX the *Klf15* gene expression was reduced in cardiac tissue. Intriguingly, KLF15 is expressed with circadian rhythmicity, being regulated by a core clock machinery that functions in the absence of external cues, including light [52], and transcriptionally coordinates rhythmic expression of multiple enzymes involved in amino acid metabolism [39]. The *Klf15*-null mice in fact exhibit reduced expression of diverse amino-acid catabolic enzymes, including alanine amino-transferase 1, proline dehydrogenase, tryptophan 2,3-dioxygenase, and 4-hydroxyphenylpyruvic acid dioxygenase [39]. KLF15 is especially crucial for diurnal rhythms of the release, uptake, and utilization of BCAAs in the heart [39]. Importantly, in the *Klf15*-null mice, cardiac pathologies are observed, such as cardiac hypertrophy and fibrosis, which suggest that KLF15 controls protein synthesis and proteolysis, likely through the regulation of the BCAA homeostasis [53,54]. Recently, Sun et al. found that a reduced KLF15 expression promoted a loss of enzymes crucial to the BCAA oxidation (e.g., BCAT2, BCKDH, and PP2Cm), resulting in accumulation of branched-chain ketoacids that, in turn, negatively impact on mitochondrial function [41]. Accordingly, the pharmacological increase of the BCAA catabolism—obtained with BT2 (3,6-dichlorobenzo[b]thiophene-2-carboxylic acid), an inhibitor of BCHDH kinase which limiting the BCKDH phosphorylation promotes its catabolic activity—significantly protected heart integrity and function in mice subjected to transverse aortic constriction [41]. In this regard, the essential amino acids and, particularly, BCAAs are known to potentiate the BCKDH catabolic activity, probably through the stimulation of mitochondrial protein phosphatase 2C (PP2Cm) and inhibition of BCKDH kinase [55]. Moreover, the oral administration of leucine activates the mouse cardiac BCKDH complex and increases cardiac levels of both leucine and α-ketoisocaproate—a metabolic intermediate of leucine oxidation [43]. Additionally, the same authors reported that the mTORC1 inhibitor rapamycin suppressed the leucine-induced BCKDH activation, even if the cardiac leucine and α-ketoisocaproate were increased as in rapamycin-untreated mice. These results suggest that mTORC1 plays a role in regulating cardiac BCAA catabolism, and are consistent with our present findings. The α5 supplementation in fact prevents mitochondrial dysfunction induced by acute DOX by promoting 1) mitochondrial biogenesis, 2) anti-ROS defense system, and 3) BCAA oxidation, with a probable production of TCA intermediates, apparently in an mTORC1-dependent manner (Figure 8).

The *Klf15*-knockdown in fact markedly reduced the mTORC1 activity in HL-1 cells and impaired the ability of α5 mixture to restore the DOX-compromised S6 phosphorylation. Similarly, the *Klf15* silencing partially prevented the α5-induced activation of BCAA catabolism—measured as BCKDH phosphorylation—both in DOX-treated and untreated cardiomyocytes. This suggests that in cardiac cells the protective effect of α5 is mediated, at least in part, by the mTORC1 activation, that depends on the KLF15 expression and, thus, on the BCAA oxidation. In accord with our findings, the KLF15 deficiency was found to inhibit the amino acid-dependent mTORC1 activation also in primary hepatocytes [56]. Moreover, we showed that in cardiac cells and tissue α5 restored to control levels Sestrin2, which is highly expressed after DOX treatment [38]. Sestrins are a family of stress-inducible proteins (Sestrin1–3) and a variety of evidence indicates that Sestrin2 is a leucine sensor for the mTORC1 pathway. Normally bound to GATOR2, which is thus inhibited under unstressed conditions, Sestrin2 is displaced from GATOR2 when leucine is present at micromolar intracellular concentrations, activating mTORC1 [37]. Mice with genetic ablation of Sestrin2 have been recently found more vulnerable to the DOX-induced cardiac dysfunction, associated with an activation of the mTORC1 pathway, suggesting that Sestrin2 is an important cardioprotective protein to preserve at balanced levels [38]. The here described ability to modulate Sestrin2 expression thus could strengthen the rationale for dietary α5 supplementation to prevent and perhaps treat the DOX cardiotoxicity.

This therapeutic rationale is supported further by the observation that our α5 formula promoted the eNOS expression and activity, in both DOX-treated and untreated cardiac cells and tissue. Probably these effects were mediated by (Ser473)-Akt phosphorylation. An extensive scientific literature demonstrates that the Akt-eNOS activation with drugs and food supplements mitigates the DOX-induced cardiac damage both in animals and humans [57,58,59,60,61,62]. Given that in endothelial cells KLF15 is a clock-dependent peripheral oscillator that controls eNOS expression [40], our present findings suggest that cardiac sensitivity to both DOX and α5 could be regulated by circadian rhythms and, thus, might need to be timely administered during the day to function at best. The role of clock proteins (e.g., Per2 and BMAL1) has been recently proposed in DOX-induced cell death and breast cancer [63,64].

A limit of the present study is the lack of *in vivo* measurements of heart physiology in mice treated with DOX and α5. Our principal purpose, however, was to first assess the ability of the novel α5 formula to prevent the cardiac mitochondrial damage induced by DOX and to characterize the molecular mechanisms involved. Our results offer to study cardiac activity with more adequate coming *in vivo* studies. A second limit of this study is the absence of exploration on the neoplastic potential of α5, although this seems to be cast doubt by increasing evidence that supplementation of amino acids—especially the BCAAs—is beneficial mainly in liver cancer, yet their role in the nutritional support of cancer patients remains to be clearly defined [65,66,67,68,69]. Similarly, we did not investigate with appropriate *in vivo* models whether the α5 supplementation can affect the antineoplastic efficacy of DOX. Nevertheless, the exposure of MCF7 breast cancer cell line to α5 mixture did not promote MCF7 cell proliferation, as assessed with two different assays (Appendix A). Similarly, the anti-proliferative effect of DOX in MCF7 cells was completely unaffected by the amino acid presence (Appendix A).

## 5. Conclusions

Overall, our findings: (1) confirm that acute DOX treatment induces mitochondrial dysfunction in both cardiac tissue and cardiomyocytes, (2) demonstrate that a designer amino acid formula, that we named α5—after five abundances, among TCA intermediates and co-factors—markedly prevents this damage, (3) suggest that eNOS and mTORC1 signaling pathways are crucially involved in the action of this protector, and (4) show that KLF15, a specific transcription factor particularly important in cardiac development and circadian regulation, plays a relevant role in controlling these signaling axis. These findings highlight that dietary α5 supplementation might be a strategy in DOX-treated patients to prevent cardiomyopathy, acting on a complex signaling network.

## Figures and Tables

**Figure 1 nutrients-12-00282-f001:**
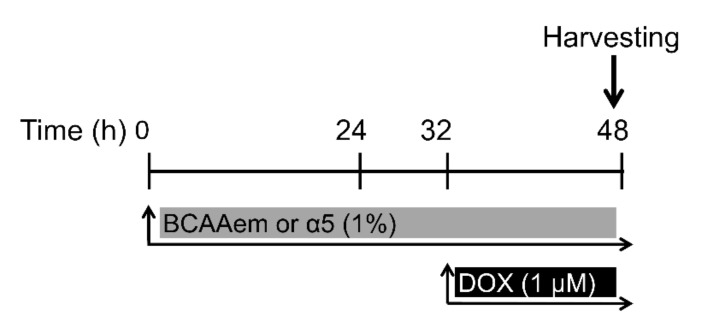
HL-1 cell treatment.

**Figure 2 nutrients-12-00282-f002:**
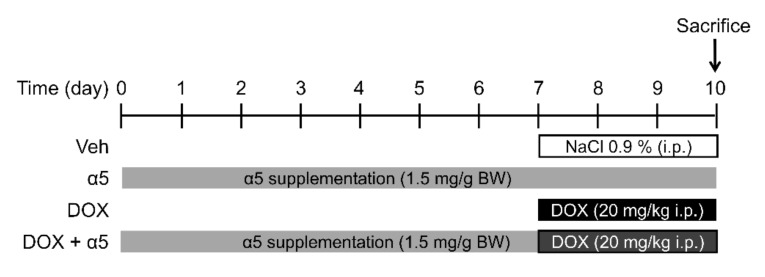
*In vivo* mouse treatment.

**Figure 3 nutrients-12-00282-f003:**
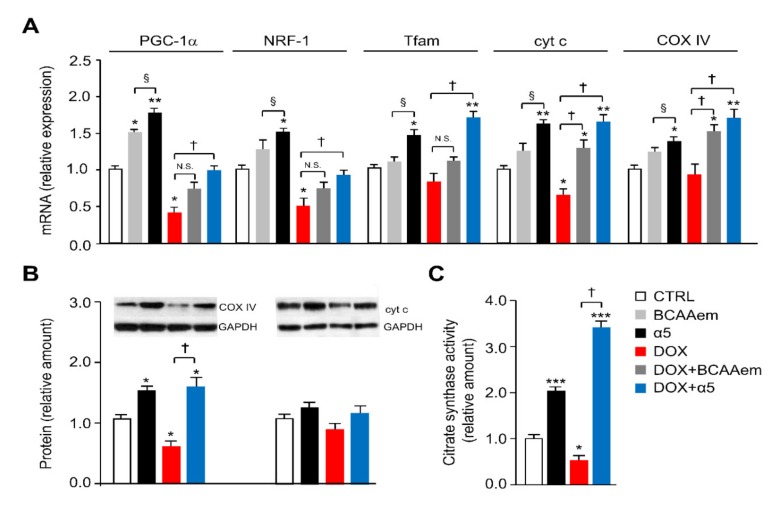
Specific amino-acid mixtures prevent mitochondrial dysfunction in HL-1 cardiomyocytes acutely exposed to DOX. (**A**) Mitochondrial biogenesis marker expression: peroxisome proliferator-activated receptor-γ coactivator 1α (PGC1-α), nuclear respiratory factor-1 (NRF1), transcription factor A (Tfam), cytochrome *c* (cyt c), and cytochrome c oxidase subunit IV (COX IV) mRNA levels were analysed by quantitative RT-PCR. Relative expression values for the untreated (CTRL) cells were taken as 1.0 (*n* = 5 experiments). (**B**) COX IV and cyt c protein levels were detected by immunoblot analysis. The relative values were determined by densitometric analysis relative to GAPDH levels; values for untreated (CTRL) cells were taken as 1.0 (*n* = 5 experiments). (**C**) Citrate synthase activity. The values were normalized to protein content (*n* = 3 experiments). ^*^
*p* < 0.05, ^**^
*p* < 0.01, and ^***^*p* < 0.001 vs. untreated cells; ^†^
*p* < 0.01 vs. DOX-treated cells; ^§^
*p* < 0.01 vs. BCAAem-treated cells. All data are presented as the mean ± SD.

**Figure 4 nutrients-12-00282-f004:**
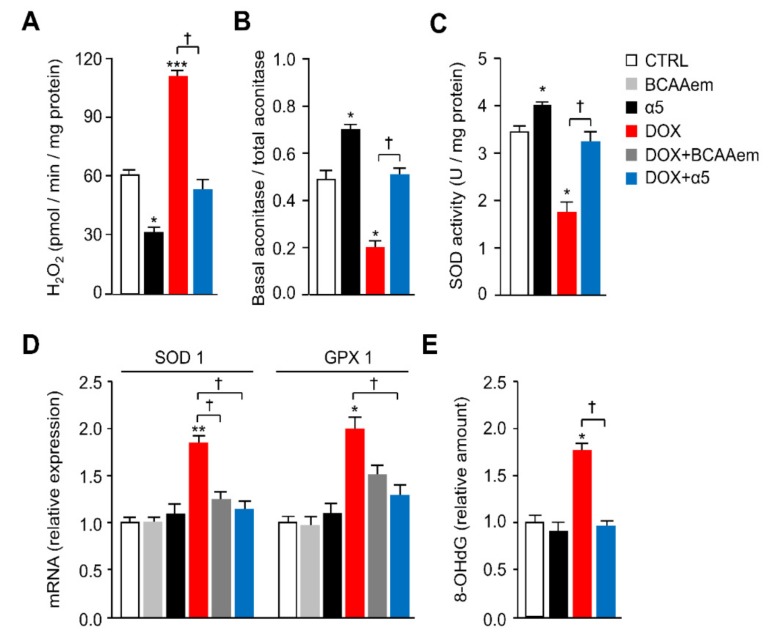
α5 Supplementation prevents DOX-induced oxidative stress in HL-1 cardiomyocytes. (**A**) Mitochondrial H_2_O_2_ release, (**B**) basal aconitase/total aconitase ratio, and (**C**) superoxide dismutase activity (SOD) in HL-1 cells (*n* = 3 experiments). (**D**) superoxide dismutase 1 (SOD1) and glutathione peroxidase 1 (GPX1) mRNA levels were analysed by quantitative RT-PCR. Relative expression values for the untreated (CTRL) cells were taken as 1.0 (*n* = 5 experiments). (**E**) Total DNA oxidative damage measured as 8-hydroxy-2’-deoxyguanosine (8-OHdG) production.

**Figure 5 nutrients-12-00282-f005:**
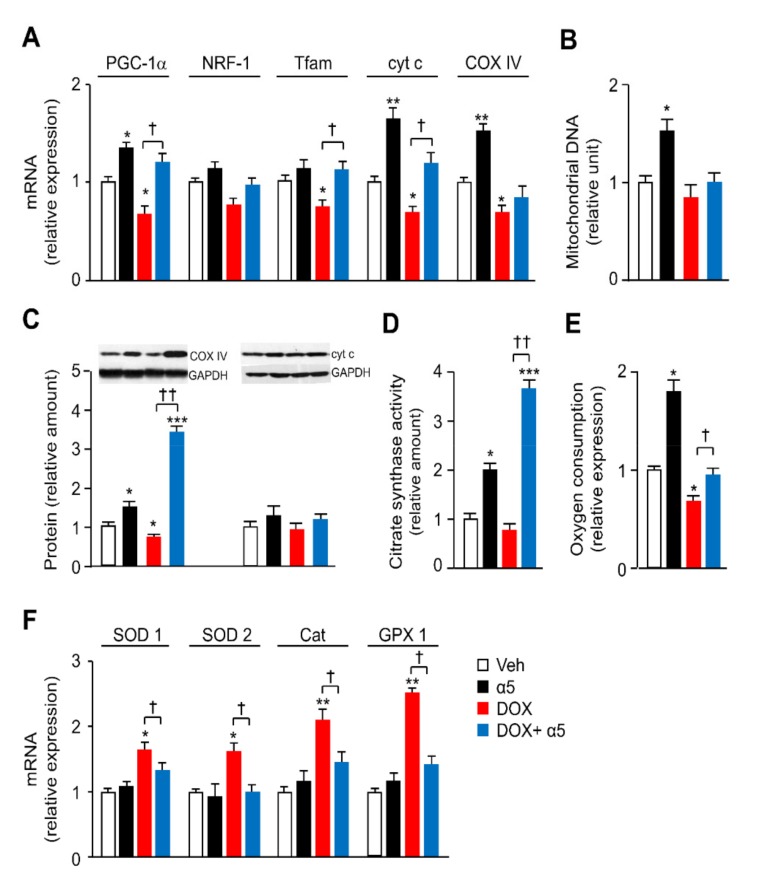
α5 formula prevents mitochondrial dysfunction in left ventricle of DOX-treated mice. (**A**,**F**) Mitochondrial biogenesis marker expression. (**A**) PGC1-α, NRF1, Tfam, cyt *c*, COX IV and (**F**) SOD1, superoxide dismutase 2 (SOD2), catalase (Cat), and GPX1 mRNA levels were analysed by quantitative RT-PCR. Relative expression values for the vehicle-treated (Veh) mice were taken as 1.0 (*n* = 5 experiments). (**B**) Mitochondrial DNA (mtDNA) amount was analyzed by quantitative RT-PCR. Relative units were expressed in comparison to those for the Veh-treated mice which were taken as 1.0 (*n* = 5 experiments). (**C**) COX IV and cyt c protein levels were determined by immunoblot analysis. The relative values were determined by densitometric analysis relative to glyceraldehyde-3-phosphate dehydrogenase (GAPDH) levels; values for Veh-treated mice were taken as 1.0 (*n* = 5 experiments). (**D**) Citrate synthase activity. The values were normalized to protein content (*n* = 3 experiments). (**E**) Basal oxygen consumption rate. Mitochondria were isolated from left ventricle of mice treated or not with DOX and α5. Oxygen consumption rates were normalized to mitochondrial protein amount (*n* = 3 experiments). ^*^
*p* < 0.05, ^**^
*p* < 0.01, and ^***^
*p* < 0.001 vs. Veh-treated mice; ^†^
*p* < 0.05 and ^††^
*p* < 0.01 vs. DOX-treated mice. All data are presented as the mean ± SD.

**Figure 6 nutrients-12-00282-f006:**
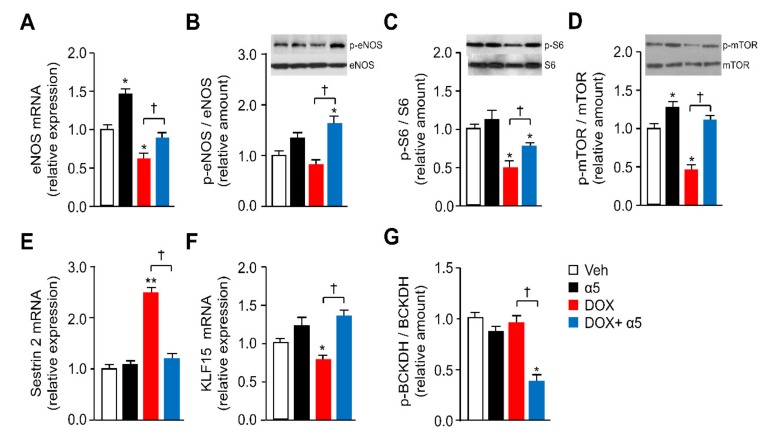
Different signaling pathways are implicated in the protective effects of α5 supplementation in DOX-treated mice. (**A**,**E**,**F**) Gene expression. (**A**) Endothelial nitric oxide synthase (eNOS), (**E**) Sestrin2, and (**F**) Krüppel-like factor 15 (KLF15) mRNA levels were analysed by quantitative RT-PCR. Relative expression values of the vehicle (Veh)-treated mice were taken as 1.0 (*n* = 5 experiments). (**B**,**C**,**D**, and **G**) Protein levels. (**B**) Phospho-eNOS, (**C**) phospho-S6, (**D**) phospho-mTOR, and (**G**) phospho-BCKDH protein levels were detected by immunoblot analysis. The relative values were densitometrically analyzed and reported as ratios to total eNOS, S6, mTOR, and BCKDH levels, respectively. Values for Veh-treated mice were taken as 1.0 (*n* = 5 experiments). ^*^
*p* < 0.05 and ^**^
*p* < 0.01 vs. Veh-treated mice; ^†^
*p* < 0.05 vs. DOX-treated mice. All data are presented as the mean ± SD.

**Figure 7 nutrients-12-00282-f007:**
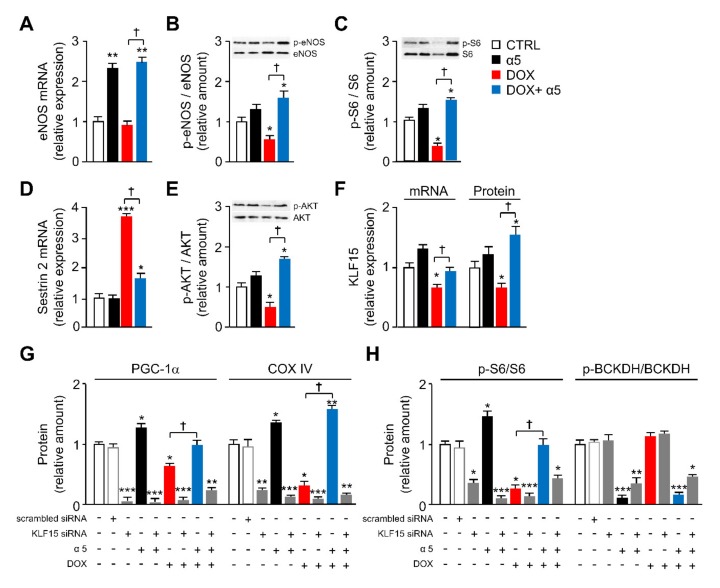
Different signaling pathways are implicated in the protective effects of α5 supplementation in DOX-treated HL-1 cardiomyocytes. (**A**, **D**, and **F**) Gene expression. (**A**) eNOS, (**D**) Sestrin2, (**F**) KLF15 mRNA levels were analysed by quantitative RT-PCR. Relative expression values for the untreated (CTRL) cells were taken as 1.0 (*n* = 5 experiments). (**B**, **C**, **E–H**) Protein levels. (**B**) Phospho-eNOS, (**C**) phospho-S6, (**E**) phospho-Akt, (**F**) KLF15, (**G**) PGC-1α and COXIV, and (**H**) phospho-S6 and phospho-BCKDH protein levels were detected by immunoblot analysis. The relative values were densitometrically analyzed and reported as ratios to total eNOS, S6, Akt, and BCKDH levels, respectively; KLF15, PGC-1α, and COXIV were normalized to GAPDH. Values for untreated HL-1 cells were taken as 1.0 (*n* = 5 experiments). (**G**) PGC1-α and COX IV protein levels were measured by immunoblot analysis in HL-1 cells transfected with either specific KLF15 siRNA or nontargeting siRNA, and treated with DOX or α5 alone or in combination. Values for untreated HL-1 cells were taken as 1.0 (*n* = 5 experiments). (**H**) S6 and BCKDH phosphorylation was measured by immunoblot analysis in HL-1 cells as in (H). ^*^
*p* < 0.05, ^**^
*p* < 0.01, and ^***^
*p* < 0.001 vs. untreated cells; ^†^
*p* < 0.05 vs. DOX-treated cells. All data are presented as the mean ± SD.

**Figure 8 nutrients-12-00282-f008:**
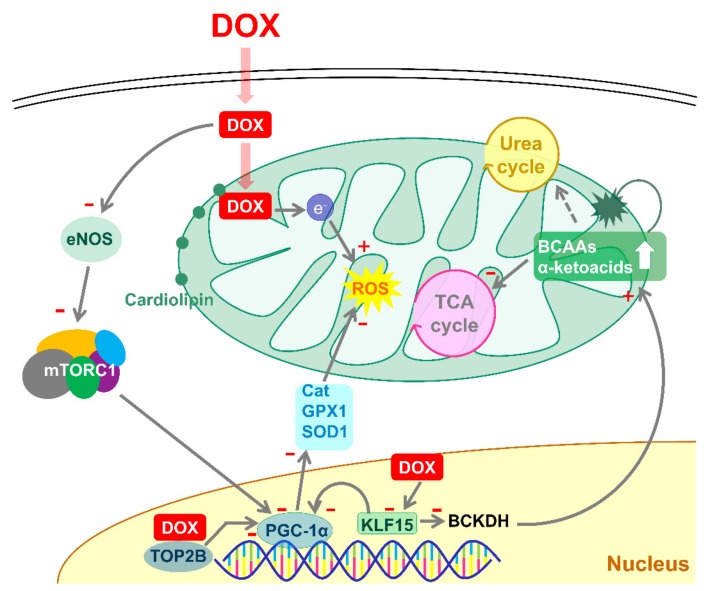
A proposed model of α5 protective actions on mitochondrial damage induced by doxorubicin (DOX) in cardiomyocytes. Plus (+) and minus (−) symbol in red colour indicate stimulation or inhibition induced by DOX and, potentially, the α5 target sites. DOX reduces mitochondrial biogenesis and function, (1) binding to topoisomerase IIβ (TOP2B), which complexes to the promoters of *Ppargc1a* and *Ppargc1b* and blocks transcription of mitochondrial genes, including ROS defense genes, (2) reducing eNOS expression and mTORC1 activity—which are important regulators of mitochondrial physiology, (3) limiting *KLF15* gene expression and presumably BCAA oxidation, with α-ketoacid and BCAA accumulation—which are toxic at high levels—beyond reduced TCA cycle intermediates and mitochondrial energy production. Mitochondrial accumulation of DOX is accompanied by increased ROS.

**Table 1 nutrients-12-00282-t001:** BCAAem and α5 composition (relative percentage).

Composition (%)	BCAAem	α5
Leucine	30.01	31.0885
Lysine	19.58	16.9030
Isoleucine	15.00	10.3628
Valine	15.00	10.3628
Threonine	8.40	7.2540
Cysteine	3.60	3.1089
Histidine	3.60	3.1089
Phenylalanine	2.40	2.0726
Methionine	1.20	1.0363
Tyrosine	0.72	0.6218
Tryptophan	0.48	2.0726
Citric acid	-	8.001
Succinic acid	-	2.00
Malic acid	-	2.00
Vitamin B1	-	0.004
Vitamin B6	-	0.0038

**Table 2 nutrients-12-00282-t002:** Primers for quantitative RT-PCR.

Gene	Sense Primer (5′-3’)	Antisense Primer (5′-3’)	PCR Product (bp)	T_a_ (°C)
*Cat*	CACTGACGAGATGGCACACTTTG	TGGAGAACCGAACGGCAATAGG	173	60
*COX IV*	TGGGACTATGACAAGAATGAGTGG	TTAGCATGGACCATTGGATACGG	113	60
*cyt c*	ATAGGGGCATGTCACCTCAAAC	GTGGTTAGCCATGACCTGAAAG	172	60
*eNOS*	AGCGGCTGGTACATGAGTTC	GATGAGGTTGTCCTGGTGTCC	116	60
*GPX1*	TCTGGGACCTCGTGGACTG	CACTTCGCACTTCTCAAACAATG	156	60
*KLF15*	ACACCAAGAGCAGCCACCTC	TGAGATCGCCGGTGCCTTGA	130	60
*mtDNA*	CCACTTCATCTTACCATTTA	ATCTGCATCTGAGTTTAATC	106	54
*NRF1*	ACAGATAGTCCTGTCTGGGGAAA	TGGTACATGCTCACAGGGATCT	99	60
*PGC-1α*	ACTATGAATCAAGCCACTACAGAC	TTCATCCCTCTTGAGCCTTTCG	148	60
*SESN2*	GCCCCTGAGAAGCTCCGCAA	GAGTTCGGCCAGGGACCAGC	129	60
*SOD1*	GGCTTCTCGTCTTGCTCTC	AACTGGTTCACCGCTTGC	153	60
*SOD2*	GCCTCCCAGACCTGCCTTAC	GTGGTACTTCTCCTCGGTGGCG	131	60
*TBP*	ACCCTTCACCAATGACTCCTATG	TGACTGCAGCAAATCGCTTGG	186	60
*Tfam*	AAGACCTCGTTCAGCATATAACATT	TTTTCCAAGCCTCATTTACAAGC	104	60
*18S*	CTGCCCTATCAACTTTCGATGGTAG	CCGTTTCTCAGGCTCCCTCTC	100	60

*Cat*, catalase; *COX IV*, cytochrome c oxidase subunit IV; *cyt c*, cytochrome c; *eNOS*, endothelial nitric oxide synthase; *GPX1*, glutathione peroxidase 1; *KLF15*, Krüppel-like factor 15; *mtDNA*, mitochondrial DNA; *NRF1*, nuclear respiratory factor 1; *PGC-1α*, peroxisome proliferator-activated receptor γ coactivator 1-α; *SESN2*, sestrin 2; *SOD1*, superoxide dismutase 1; *SOD2*, superoxide dismutase 2; *TBP*, TATA-binding protein; *Tfam*, mitochondrial transcription factor A; *18S*, 18S ribosomal RNA.

**Table 3 nutrients-12-00282-t003:** Body weight, heart weight, food intake, and water consumption.

	CTRL	α5	DOX	DOX + α5
Body weight (g) Heart weight (g) Food intake (g) Water intake (g)	24.84 ± 1.3 0.12 ± 0.01 5.2 ± 0.41 6.1 ± 0.7	24.46 ± 2 0.11 ± 0.02 5.34 ± 0.7 8.24 ± 0.88 *	20.75 ± 0.82 * 0.085 ± 0.02 * 3.73 ± 0.6* 6.13 ± 1.0	20.46 ± 1.04 * 0.09 ± 0.02 * 3.59 ± 0.47 * 7.07 ± 1.3

Measurements were done in 10 mice per group. Values represent mean ± D.S. * *p* < 0.05 vs. controls (i.e., saline-injected mice).

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
