# Peer review of "A Special Amino-Acid Formula Tailored to Boosting Cell Respiration Prevents Mitochondrial Dysfunction and Oxidative Stress Caused by Doxorubicin in Mouse Cardiomyocytes"

_nutrients, 2020, doi:10.3390/nu12020282_

Round 1

Reviewer 1 Report

The authors tat examinedα 5 (the essential amino acids, tricarboxylic acid cycle precursors) function on preventing side effect of doxorubicin (DOX). DOX treatment also can trigger cardiotoxicity from mitochondrial damage. They found that α5 mixture prevents the DOX-dependent mitochondrial damage and oxidative stress better than the previous BCAAem, which regulated the KLF15/eNOS/mTORC1 signaling axis. It is interesting. Some comments should be considered during publish.

Major comment:

In the DOX treatment can induce ROS production in HL-1 cell and mouse system should be assayed. Then, inα5 treatment group should monitoring ROS level within cells for make sure oxidative stress in reducing in cell or mouse.

During treatment with DOX or α5 also need to monitoring mitochondrial function such as mitochondrial network structure by staining with mitochondrial dye JC-1 or Mitotracker.

In conclusion has mentioned “TCA intermediates and co-factors—markedly prevents this damage”, why directly test those compound function in cells?

Minor comments:

In figure 5, in figure legend should mention positive (+) and negative (-) signaling in the scheme.

Stress signal and regulation should be enhanced in Introduction such as ROS signal and catalase or SOD.

Reviewer 2 Report

The manuscript submitted by Tedesco et al. studied the toxicity of doxorubicin (DOX) and how to use a new formula to prevent cardiotoxicity. They proposed a mitochondrial pathway in which multiple factors involved, eNOS/mTORC1/KLF15. The study was well designed and performed. Although it has shortcomings, overall it is a good research.

I have some specific concerns/points that shared with authors and expect to see some improvement in the revised manuscript.

Authors used a combination of essential amino acids, tricarboxylic acid cycle precursors and co-factors (named a5) to reverse the mitochondrial damages. My question is whether the levels of these metabolites are also changed?  Since the authors studied the mTORC1 pathway, I would like to know the regulation of mTORC2 pathway. Authors usually used mRNA levels to indicate the protein changes, which is not considered accurate. There is always a post-translational modification effect. How does mTORC1 activity get manipulated, through phosphorylation on active sites, or the assembly/disassembly of the complexes? In Figure 4, authors studied the effects of siKLF15, which is good. How about knocking-down mTORC1/eNOS1 or other regulators in this pathway? Minor things. It seems that the Suppl Figures embedded in the manuscript is mislabeled. They are missing S. For instance, on page 8, it should be Figure S1. instead of Figure 1. 

Round 2

Reviewer 1 Report

No one.

Reviewer 2 Report

The authors have addressed my concerns. I support to publish this manuscript.

Although I think on page 18, line 600, 601; it should be figure S5A, S5B, not S3A, S3B.